# Conflict-Aware Adversarial Training

## Abstract

Adversarial training is the most effective method to obtain adversarial robustness for deep neural networks by directly involving adversarial samples in the training procedure. To obtain an accurate and robust model, the weighted-average method is applied to optimize standard loss and adversarial loss simultaneously. In this paper, we argue that the weighted-average method does not provide the best tradeoff for standard performance and adversarial robustness. We argue that the failure of the weighted-average method is due to the conflict between gradients derived from standard and adversarial loss, and further demonstrate such a conflict increases with attack budget theoretically and practically. To alleviate this problem, we propose a new trade-off paradigm for adversarial training with a conflict-aware factor for the convex combination of standard and adversarial loss, named **Conflict-Aware Adversarial Training (CA-AT)**. Comprehensive experimental results show that CA-AT consistently offers a superior trade-off between standard performance and adversarial robustness under the settings of adversarial training from scratch and parameter-efficient finetuning.

## 1 Introduction

Deep learning models have achieved exemplary performance across diverse application domains (He et al., 2017; Vaswani et al., 2017; Ouyang et al., 2022; Rombach et al., 2022; Radford et al., 2021). However, they remain vulnerable to adversarial samples produced by adversarial attacks (Goodfellow et al., 2014; Liu et al., 2016; Moosavi-Dezfooli et al., 2016). Deep learning models can easily be fooled into making mistakes by adding an imperceptible noise produced by adversarial attacks to the standard sample. To solve this problem, many methods have been proposed to improve the robustness against adversarial samples (Cai et al., 2018; Chakraborty et al., 2018; Madry et al., 2018), among which **adversarial training (AT)** has been proven to be the most effective strategy (Madry et al., 2018; Athalye et al., 2018; Qian et al., 2022; Bai et al., 2021). Specifically, AT aims to enhance model robustness by directly involving adversarial samples during training. They used adversarial examples to construct the adversarial loss functions for parameter optimization. The adversarial loss can be formulated as a min-max optimization objective, where the adversarial samples are generated by the inner maximization, and the model parameters are optimized by the outer minimization to reduce the empirical risk for adversarial samples.

The trade-off between standard and adversarial accuracy is a key factor for the real-world applications of AT (Tsipras et al., 2018; Balaji et al., 2019; Yang et al., 2020b; Stutz et al., 2019; Zhang et al., 2019). Although AT can improve robustness against adversarial samples, it also undermines the performance on standard samples. Existing AT methods (Madry et al., 2018; Cai et al., 2018; Zhang et al., 2019; Wang et al., 2019) design a hybrid loss by combining standard loss and an adversarial loss linearly, where the linear coefficient typically serves as the trade-off factor.

In this paper, we argue that linearly weighted-average method for AT, as well as the **Vanilla AT**, cannot achieve a 'near-optimal' trade-off. In other words, it fails to approximately achieve the Pareto optimal points on the Pareto front of standard and adversarial accuracies. We find that the conflict between the parameter gradient derived from standard loss (**standard gradient**) and the one derived from adversarial loss (**adversarial gradient**) is the main source of this failure. Such a gradient conflict causes the model parameter to be stuck in undesirable local optimal points, and it becomes more severe with the increase of

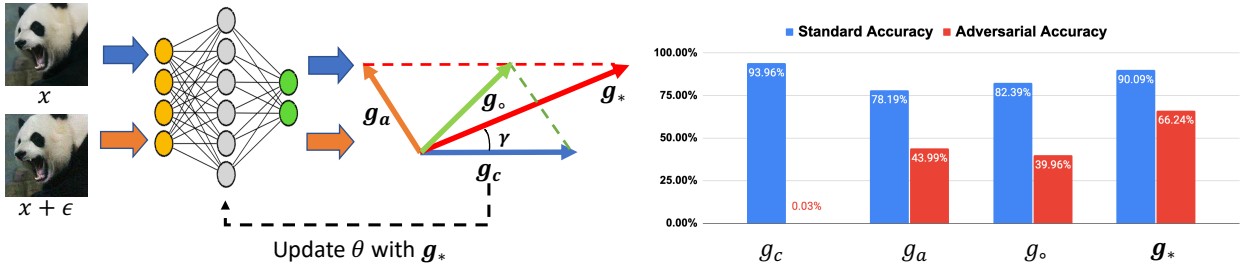

Figure 1: The key motivation of CA-AT aims to solve the conflict between clean gradient $g_c$ and adversarial gradient $g_a$. Unlike the existing weighted-averaged method optimizing model parameter $\theta$ by $g_\circ$ as the average of $g_c$ and $g_a$ (Vanilla AT), CA-AT utilizes $g_*$ for parameter optimization by gradient projection based on a new trade-off factor $\phi$. The bar chart on the right side illustrates that the model optimized by $g_*$ (highlighted as the boldface) can achieve better standard accuracy (blue bar) and adversarial accuracy (red bar) compared to models optimized by $g_\circ$. The results of the bar chart on the right are produced by training a ResNet18 on CIFAR10 against the PGD (Madry et al., 2017) attack.

adversarial attack budget. In addition, to obtain adversarial robustness, linearly weighted-average method usually sacrifices too much performance on standard samples, which hinders AT from real-world applications.

To solve the problems mentioned above, we propose **Conflict-Aware Adversarial Training (CA-AT)** to mitigate the conflict during adversarial training. Inspired by gradient surgery (Yu et al., 2020) in multi-task learning, CA-AT utilizes a new trade-off factor defined as the angle between the standard and adversarial gradients. As depicted in Fig. 1, if the angle is larger than the pre-defined trade-off factor $\gamma$, CA-AT will project the adversarial gradient onto the 'cone' around the standard gradient constructed based on the pre-defined trade-off factor; otherwise, it will use the standard gradient to optimize the model parameter $\theta$ directly. Compared to the linearly weighted-average AT with a fixed trade-off factor, CA-AT can boost both standard and adversarial accuracy. Our primary contributions are summarized as follows:

1. We shed light on the existence of conflict between standard and adversarial gradient which causes a sub-optimal trade-off between standard and adversarial accuracy in AT, when we optimize standard and adversarial loss in weighted-average paradigm by a fixed trade-off factor.

2. To alleviate the gradient conflict in AT, we propose a new paradigm called Conflict-Aware Adversarial Training (CA-AT). It achieves a better trade-off between standard and adversarial accuracy compared to Vanilla AT.

3. Through comprehensive experiments across a wide range of settings, we demonstrate CA-AT consistently improves the trade-off between standard and adversarial accuracy in the context of training from scratch and parameter-efficient finetuning (PEFT), across diverse adversarial loss functions, adversarial attack types, model architectures, and datasets.

## 2 Related Works

**Adversarial Training.** Adversarial training (AT) is now broadly considered as the most effective method to achieve adversarial robustness for deep learning models (Qian et al., 2022; Singh et al., 2024). The key idea of AT is to involve adversarial samples during the training process. Existing works for AT can be mainly grouped into regularization-driven and strategy-driven. For regularization-driven AT methods, the goal is to design an appropriate loss function for adversarial samples, such as cross-entropy (Madry et al., 2017), logits pairing (CLP) (Kannan et al., 2018), and TRADES (Zhang et al., 2019). On the other hand, strategy-driven AT methods focus on improving adversarial robustness by designing appropriate training strategies. For example, ensemble AT (Tramèr et al., 2017; Yang et al., 2020a) alleviates the sharp parameter curvature by utilizing adversarial examples generated from different target models, curriculum AT (Cai et al., 2018) gains adversarial robustness progressively by learning from easy adversarial samples to hard adversarial samples,

and adaptive AT (Ding et al., 2018; Cheng et al., 2020; Jia et al., 2022) improves adversarial robustness by adjusting the attack intensity and attack methods. With the development of large-scale pretrained models (Kolesnikov et al., 2020; Dong et al., 2020), (Jia et al., 2024; Hua et al., 2023) demonstrates the superiority of adversarial PEFT of robust pretrained models, compared to adversarial training from scratch.

However, strategy-based AT methods need to involve additional attack methods or target models in the training process, which will increase the time and space complexity when we apply them. CA-AT can improve both standard and adversarial performance without any increasing cost of training time and computing resources.

**Gradients Operation.** Gradients Operation, also known as gradient surgery (Yu et al., 2020), aims to improve model performance by directly operating the parameter gradient during training. It was first presented in the area of multi-task learning to alleviate the gradient conflict between loss functions designed for different tasks. The conflict can be measured by cosine similarity (Yu et al., 2020) or Euclidean distance (Liu et al., 2021a) between the gradients derived from different loss functions. Besides, multi-task learning, (Mansilla et al., 2021) incorporates gradient operation to encourage gradient agreement among different source domains, enhancing the model's generalization ability to the unseen domain, and (Chaudhry et al., 2018; Yang et al., 2023) alleviate the forgetting issue in continual learning by projecting the gradients from the current task to the orthogonal direction of gradients derived from the previous task.

We are the first work to observe the gradient conflict between standard and adversarial loss during AT and further reveal its relation to adversarial attack budget. Moreover, we propose a new trade-off paradigm specifically designed for AT based on gradient operation. It can achieve a better trade-off compared to Vanilla AT and guarantee the standard performance well.

## 3 The Phenomenon of Gradient Conflict in AT

In this section, we will discuss the occurrence of gradient conflict in AT via a synthetic dataset and real-world datasets such as CIFAR10 and CIFAR100. Additionally, we demonstrate such a conflict will become more serious with the increase of the attack budget theoretically and practically.

### 3.1 Preliminaries & Notations

Considering a set of images, each image $x \in \mathbb{R}^d$ and its label $y \in \mathbb{R}^l$ is drawn i.i.d. from distribution $\mathcal{D}$. The classifier $f : \mathbb{R}^d \to \mathbb{R}^l$ parameterized by $\theta$ aims to map an input image to the probabilities of the classification task. The objective of AT is to ensure that $f$ does not only perform well on $x$, but also manifests robustness against adversarial perturbation $\epsilon$ bounded by attack budget $\delta$ as $\|\epsilon\|_p \leq \delta$, where $p$ determinates the $L_p$ norm constraint on the perturbations $\epsilon$ commonly taking on the values of $\infty$ or 2. The perturbation $\epsilon$ can be defined as $\epsilon = \arg\max_{\|\epsilon\|_p \leq \delta} \mathcal{L}(x + \epsilon, y; \theta)$, which can be approximated by gradient-based adversarial attacks such as PGD. Throughout the remaining part of this paper, we refer to $x$ as the standard sample and $x + \epsilon$ as the adversarial sample.

We define clean loss $\mathcal{L}_c = \mathcal{L}(x, y; \theta)$ and adversarial loss $\mathcal{L}_a = \mathcal{L}(x + \epsilon, y; \theta)$, respectively. $\mathcal{L}$ is the loss function for classification task (e.g. cross-entropy). As shown in Eq. (1), the goal of adversarial training is to obtain the parameter $\theta$ that can be both accurate and robust.

$$\min_\theta (\mathbb{E}_{(x,y)\sim\mathcal{D}}[\mathcal{L}_c], \mathbb{E}_{(x,y)\sim\mathcal{D}}[\mathcal{L}_a]) \tag{1}$$

For vanilla AT, as mentioned in Section 2, optimizing a hybrid loss containing standard loss $\mathcal{L}_c$ and adversarial loss $\mathcal{L}_a$ is a widely-used method for solving Eq. (1). As shown in Eq. (2), existing works (Wang et al., 2019; Zhang et al., 2019; Kannan et al., 2018) construct such a hybrid loss by using a linear-weighted approach for $\mathcal{L}_c$ and $\mathcal{L}_a$.

$$\min_\theta \mathbb{E}_{(x,y)\sim\mathcal{D}}[\lambda\mathcal{L}_a + (1 - \lambda)\mathcal{L}_c], \tag{2}$$

where $\lambda \in [0, 1]$ serves as a fixed hyper-parameter for the trade-off between $\mathcal{L}_c$ and $\mathcal{L}_a$. Refer to Fig. 1, the optimization process of Eq. (2) can be described as utilizing $g_o = (1 - \lambda)g_c + \lambda g_a$ to update $\theta$ at each optimization step, where $g_c = \frac{\partial \mathcal{L}_c}{\partial \theta}$ and $g_a = \frac{\partial \mathcal{L}_a}{\partial \theta}$ represent standard and adversarial gradients, respectively.

To measure how well we can solve Eq. (1), we define a metric $\mu = ||g_c||_2 \cdot ||g_a||_2 \cdot (1 - \cos(g_c, g_a))$. The basic motivation for the consideration of $\mu$ is that it should combine three kinds of signals during AT simultaneously: **(1)** $||g_c||_2$ reflects the convergence of clean loss $\mathcal{L}_c$, **(2)** $||g_a||_2$ reflects the convergence of adversarial loss $\mathcal{L}_a$, and **(3)** $(1 - \cos(g_c, g_a))$ reflects the directional conflict between $g_c$ and $g_a$. Based on **(1)**, **(2)**, and **(3)**, a small $\mu$ implies that both $\mathcal{L}_c$ and $\mathcal{L}_a$ have converged well while reaching a consensus on the optimization direction for the next step.

## 3.2 Theoretical & Experimental Support for Motivation

We introduce Theorem 1 that demonstrates $\mu$ can be bounded by the input dimension $d$ and perturbation budget $\delta$ in AT.

**Theorem 1.** *Consider the gradient conflict $\mu = ||g_c||_2 \cdot ||g_a||_2 \cdot (1 - \cos(g_c, g_a))$ and suppose that the input $x$ is a d-dimensional vector.*

1. *Given the $L_2$ restriction for $\epsilon$ as $||\epsilon||_2 \le \delta$, we have $\mu \le \mathcal{O}(\delta^2)$.*

2. *Given the $L_\infty$ restriction for $\epsilon$ as $||\epsilon||_\infty \le \delta$, we have $\mu \le \mathcal{O}(d^2\delta^2)$.*

The intuitive understanding of Theorem 1 is that with the increasing attack budget $\delta$, the adversarial samples in AT will move further from the distribution $\mathcal{D}$ of standard samples. The conflict between $g_a$ and $g_c$ will become more serious, and $L_a$ and $L_c$ will be hard to converge. Therefore, the upper bound of $\mu$ will become larger. The proof of Theorem 1 will be shown in the appendix.

**Synthetic Experiment.** In order to show the implications of Theorem 1 empirically, we introduce the synthetic experiment as a binary classification task by selecting digit one and digit two from MNIST with a resolution of $32 \times 32$, and train a logistic regression model parameterized by $w \in \mathbb{R}^{(32 \times 32) \times 2}$ via BCE loss by vanilla AT for 20 epochs, where $\epsilon$ is contained by its $L_\infty$ norm as $||\epsilon||_\infty \le \delta$, and $\lambda = 0.5$ serves as the trade-off factor between standard and adversarial loss. Compared to the experiments on real-world datasets, this synthetic experiment offers a distinct advantage in terms of the ability to analytically solve the inner maximization. For real-world datasets, only numerical solutions can be derived using gradient-based attacks (e.g. PGD) during AT. These numerical solutions sometimes are not promising due to gradient masking (Athalye et al., 2018; Papernot et al., 2017). On the contrary, our synthetic experiments can ensure a high-quality solution for inner maximization, eliminating the potential effect of experimental results caused by some uncertainties such as gradient masking.

Under the circumstance of a simple logistic regression model with analytical solution for inner maximization, the hybrid loss for Vanilla AT can be presented as Eq. (3), where exp() denotes the exponential function. The details of getting the analytical solution for inner maximization will be presented in the appendix.

$$\min_\theta \mathbb{E}_{(x,y)\sim\mathcal{D}}[\lambda \log(1 + \exp(-y \cdot (w^T x + b) + \delta||w||_1))$$
$$+ (1 - \lambda) \log(1 + \exp(-y \cdot (w^T x + b)))] \tag{3}$$

Fig. 2 illustrates the results of this synthetic experiment. By TSNE, Fig. 2a visualizes the distributions of $g_a$ and $g_c$ for different training samples in the last training epoch. With the increase of attack budget $\delta$, these two distributions are progressively fragmented, meaning $g_a$ and $g_c$ become more different.

For Fig. 2a, it is the tSNE visualization depicting the distributions of $g_a$ and $g_c$ for different training samples across varying $\delta$. Particularly, the distributions of $g_a$ and $g_c$ begin to segregate more distinctly as $\delta$ becomes larger, concomitant with the increasing gradient conflict $\mu$. Furthermore, the bar chart Fig. 2b reveals a decline in both standard and adversarial accuracies with increasing $\delta$ and $\mu$. This trend indicates that the larger gradient conflict can harm the model's performances on both standard and adversarial accuracies. The

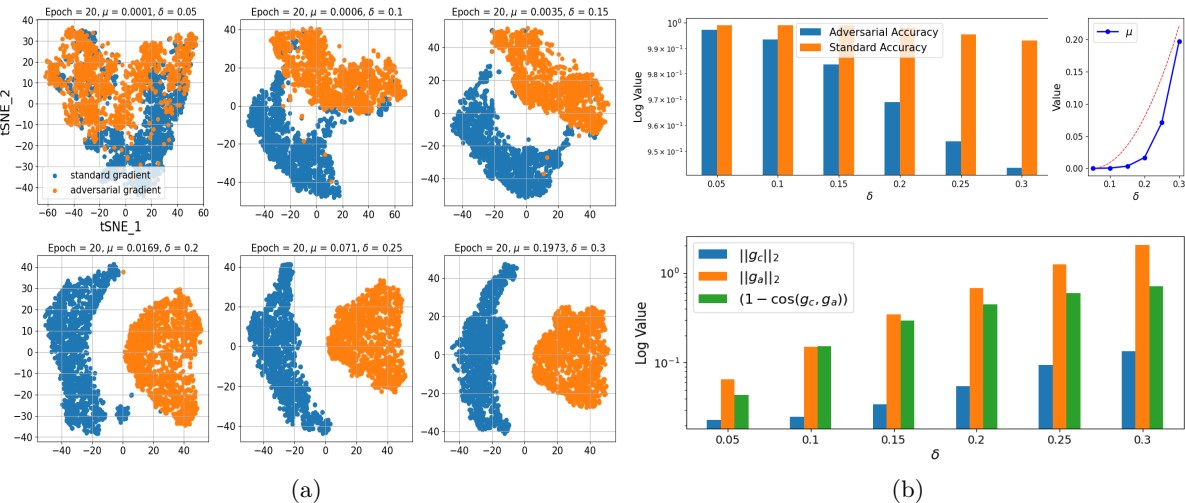

(a)  (b)

Figure 2: The experimental results of conducting Vanilla AT with $\lambda = 0.5$ for a binary classification task on our MNIST-crafted data. In **Fig. 2a**, each subfigure is the tSNE (Hinton & Roweis, 2002) visualization displaying the distribution of adversarial gradients ($g_a$) and standard gradients ($g_c$) for various training samples at the final epoch with different attack budgets ($\delta = [0.05, 0.1, 0.15, 0.2, 0.25, 0.3]$). In **Fig. 2b**, the upper bar chart shows the standard and adversarial accuracy on testing set with different $\delta$ similar to Fig. 2a. The upper left line chart shows the relation between the $\mu = ||g_c||_2 \cdot ||g_a||_2 \cdot (1 - \cos(g_c, g_a))$ and $\delta$, where the red line is the theoretical upper bound presented in Theorem 1. For decomposing $\mu$, lower bar chart shows the relation between $\delta$ and $||g_c||_2/||g_a||_2/(1 - \cos(g_a, g_c))$, respectively.

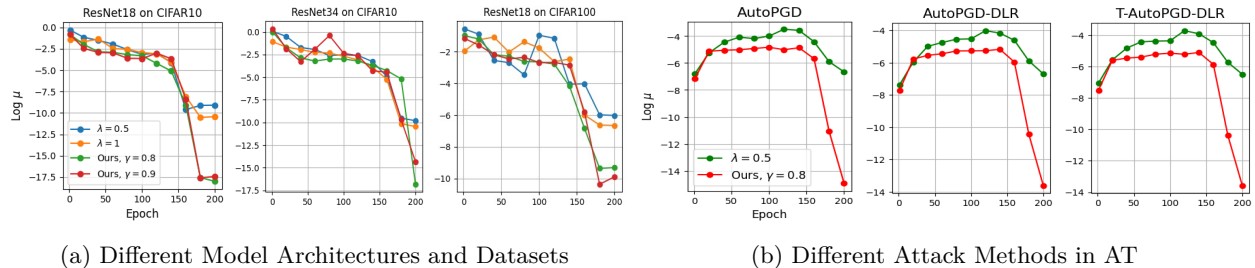

(a) Different Model Architectures and Datasets  (b) Different Attack Methods in AT

Figure 3: Results of gradient conflict metric $\mu$ on real-world datasets. Fig. 3a illustrates the results of $\mu$ among different real-world datasets (CIFAR10/CIFAR100) and model architectures (ResNet18/ResNet34), where the attack method used in AT is PGD. Fig. 3b shows the results of $\mu$ for different attack methods (AutoPGD/AutoPGD-DLR/T-AutoPGD-DLR) during AT, conducted on CIFAR10 with ResNet18.

subfigure on the right side of Fig. 2b shows an almost quadratic growth relationship between $\mu$ and $\delta$, the red line is the theoretical upper bound derived from Theorem 1, demonstrating the effectiveness of Theorem 1 empirically.

**Experiments on Real-world Datasets.** Beyond the synthetic experiment, we also conduct experiments on real-world datasets such as CIFAR10/CIFAR100, and we also observe the gradient conflict during AT. Fig. 3 shows that such a conflict exists varying from different datasets, model architectures, and attack methods used in AT, and our method ($\gamma = 0.8$, $\gamma = 0.9$), which will be introduced in the next section, can consistently alleviate the conflict compared to the Vanilla AT ($\lambda = 0.5$, $\lambda = 1$). For the fluctuation of the red line (Ours, $\gamma = 0.9$) between epochs 60–90 in the middle figure of Fig. 3a, it can be attributed to the learning rate schedule. During these epochs, the one-cycle learning rate schedule we used involves a high learning rate, which can result in increased instability and thus larger fluctuations for the gradient conflict $\mu$.

## 4 Methodology

---

**Algorithm 1** CA-AT

---

**Input:** Training dataset $D$, Loss function $\mathcal{L}$, Perturbation budget $\delta$, Training epochs $N$, Initial model parameter $\theta_1$, Projection margin threshold $\gamma$, learning rate $lr$

**Output:** Trained model parameter $\theta_{N+1}$

1: **for** $t = 1$ to $N$ **do**
2:     **for** each batch $B$ in $D$ **do**
3:         $\mathcal{L}_{\text{c}} = \frac{1}{|B|} \sum_{(x,y) \in B} \mathcal{L}(x, y; \theta_t)$
4:         $\mathcal{L}_{\text{a}} = \frac{1}{|B|} \sum_{(x,y) \in B} \max_{||\epsilon||_\infty \leq \delta} \mathcal{L}(x + \epsilon, y; \theta_t)$
5:         $g_{\text{c}}, g_{\text{a}} = \nabla_{\theta_t} \mathcal{L}_{\text{c}}, \nabla_{\theta_t} \mathcal{L}_{\text{a}}$
6:         $\phi = \cos(g_{\text{c}}, g_{\text{a}})$
7:         **if** $\phi < \gamma$ **then**
8:             $g_* = g_{\text{a}} + \frac{||g_{\text{a}}||_2 (\gamma\sqrt{1-\phi^2} - \phi\sqrt{1-\gamma^2})}{||g_{\text{c}}||_2 \sqrt{1-\gamma^2}} g_c$
9:         **else**
10:            $g_* = g_{\text{c}}$
11:         **end if**
12:         $\theta_t = \theta_t - lr * g_*$
13:     **end for**
14:     $\theta_{t+1} = \theta_t$
15: **end for**

---

As we mentioned in Section 3, the trade-off between standard and adversarial accuracy is profoundly influenced by the gradient conflict $\mu$ (Fig. 2). The vanilla AT, which employs a linear trade-off factor $\lambda$ to combine clean and adversarial loss (as seen in Eq. (2)), does not adequately address the issue of gradient conflict.

Based on this observation, we introduce Conflict-aware Adversarial Training (CA-AT) as a new trade-off paradigm for AT. The motivation of CA-AT is that the gradient conflict in AT can be alleviated by generally conducting operations on the adversarial gradient $g_{\text{a}}$ and the standard gradient $g_{\text{c}}$ during the training process, and such an operation should guarantee the standard accuracy because its priority is higher than the adversarial accuracy. Inspired by existing works related to gradient operation Yu et al. (2020); Liu et al. (2021a); Chaudhry et al. (2018); Mansilla et al. (2021), CA-AT employs a pre-defined trade-off factor $\gamma$ as the goal of cosine similarity between $g_{\text{c}}$ and $g_{\text{a}}$. In each iteration, instead of updating parameter $\theta$ by linearly weighted-averaged gradient $g_{\text{o}}$, CA-AT utilizes $g_*$ to update $\theta$ as Eq. (4)

$$g_* = \begin{cases} g_{\text{a}} + \frac{||g_{\text{a}}||_2 (\gamma\sqrt{1-\phi^2} - \phi\sqrt{1-\gamma^2})}{||g_{\text{c}}||_2 \sqrt{1-\gamma^2}} g_{\text{c}}, & \phi \leq \gamma \\ g_{\text{c}}, & \phi > \gamma \end{cases} \tag{4}$$

where $\phi = \cos(g_{\text{a}}, g_{\text{c}})$ is the cosine similarity between standard gradient $g_{\text{c}}$ and adversarial gradient $g_{\text{a}}$. The intuitive explanation of Eq. (4) is depicted in Fig. 1. For each optimization iteration, if $\phi$ is less than $\gamma$, then $g_*$ is produced by projecting $g_{\text{a}}$ onto the cone of $g_{\text{c}}$ at an angle $\arccos(\gamma)$. If $\phi > \gamma$, we will use the standard gradient $g_{\text{c}}$ to optimize $\theta$, because we need to guarantee standard accuracy when the conflict is not quite serious.

The mechanism behind Eq. (4) is straightforward. It mitigates the gradient conflict in AT by ensuring that $g_{\text{c}}$ is consistently projected in a direction close to $g_{\text{a}}$. Considering an extreme case that $g_{\text{c}}$ and $g_{\text{a}}$ are diametrically opposite ($g_{\text{a}} = -g_{\text{c}}$), in such a scenario, if we produce the gradient by Vanilla AT as $g_{\text{o}} = g_{\text{c}} + g_{\text{c}}$, $g_{\text{o}}$ will be a zero vector and the optimization process will be stuck. On the other hand, $g_*$ will align closely to $g_{\text{c}}$ within $\gamma$, avoiding $\theta$ to be stuck in a suboptimal point.

Furthermore, under the condition of $\phi \leq \gamma$, we find that CA-AT can also be viewed as a convex combination for standard and adversarial loss with a conflict-aware trade-off factor $\lambda^*$ as $\mathcal{L} = \mathcal{L}_{\text{c}} + \lambda^* \mathcal{L}_{\text{a}}$, where $\lambda^* = \frac{||g_{\text{a}}||_2 (\gamma\sqrt{1-\phi^2} - \phi\sqrt{1-\gamma^2})}{||g_{\text{c}}||_2 \sqrt{1-\gamma^2}}$. Intuitively, $\lambda^*$ increases with the decreasing of $\phi$, which means we lay more emphasis

on the standard loss when the conflict becomes more serious, and the hyperparameter $\gamma$ here serves a role of temperature to control the intensity of changing to $\lambda^*$.

The pseudo-code of the CA-AT is shown as Algorithm 1. In each training batch $B$, we calculate both standard loss $\mathcal{L}_c$ and adversarial loss $\mathcal{L}_a$. By evaluating and adjusting the alignment between standard gradient $g_c$ and adversarial gradient $g_a$, the algorithm ensures the model not only performs well via standard samples but also maintains robustness against designed perturbations. This adjustment is made by modifying the adversarial gradient $g_a$ to better align with the standard gradient $g_c$ based on the projection margin threshold $\gamma$, where $g_*$ is produced to optimize the model parameter $\theta_t$ in each round $t$.

## 5 Experimental Results & Analysis

In this section, we demonstrate the effectiveness of CA-AT for achieving better trade-off results compared to Vanilla AT. We conduct experiments on adversarial training from scratch and adversarial PEFT among various datasets and model architectures. Besides, motivated by Theorem 1, we evaluate CA-AT by involving adversarial samples with a larger budget in training. Experimental results show that CA-AT can boost the model's robustness by handling adversarial samples with a larger budget, while Vanilla AT fails.

### 5.1 Experimental Setup

**Datasets and Models.** We evaluate our proposed method on various image classification datasets including CIFIAR10 (Krizhevsky et al., 2009), CIFIAR100 (Krizhevsky et al., 2009), CUB-Bird (Wah et al., 2011), and StanfordDogs (Khosla et al., 2011). The model architectures we utilized to train from scratch on CIFAR10 and CIFAR100 are ResNet18, ResNet34 (He et al., 2016), and WideResNet28-10 (WRN-28-10) (Zagoruyko & Komodakis, 2016). We set the value of running mean and running variance in each Batch Normalization block into false as a trick to boost adversarial robustness (Wang et al., 2022; Walter et al., 2022). For experiments on PEFT, we fine-tune Swin Transformer (Swin-T) (Liu et al., 2021b) and Vision Transformer (ViT) (Dosovitskiy et al., 2020) by using Adapter (Pfeiffer et al., 2020; 2021), which fine-tunes the large pretrained model by inserting a small trainable module into each block. Such a module adapts the internal representations for specific tasks without altering the majority of the pretrained model's parameters. Both Swin-T and ViT are pretrained adversarially (Dong et al., 2020) on ImageNet. For the experiments on ResNet, we set the resolution of input data as $32 \times 32$, and use resolution as $224 \times 224$ for the PEFT experiments on Swin-T and ViT.

**Hyper-parameters for AT.** For adversarial training from scratch, we use the PGD attack with $\delta = 8/255$ with step size $2/255$ and step number 10. For the optimizer, we use SGD with momentum as 0.9 and the initial learning rate as 0.4. We use the one-cycle learning rate policy (Smith & Topin, 2019) as the dynamic adjustment method for the learning rate within 200 epochs. The details of hyperparameter setup for adversarial PEFT will be shown in Appendix. Generally, we use a sequence of operations as random crop, random horizontal flip, and random rotation for data augmentation. For a fair comparison, we maintain the same hyper-parameters across experiments for vanilla AT and CA-AT on both adversarial training from scratch and PEFT.

**Evaluation.** We evaluate adversarial robustness by reporting the accuracies against extensive adversarial attacks constrained by $L_\infty$ and $L_2$. For attacks bounded by $L_\infty$ norm, we selected most representative methods including PGD (Madry et al., 2018), AutoPGD (Croce, 2020), FGSM (Goodfellow et al., 2014), MIFGSM (Dong et al., 2018), FAB (Croce & Hein, 2020), and AutoAttack (Croce, 2020). Besides, we also conducted the targeted adversarial attacks, where they are denoted as a 'T-' as the prefix (e.g. T-AutoPGD). For all the targeted adversarial attacks, we set the number of classes as 10. Attacks bounded by $L_2$ norm are denoted as '-L2' in suffix (e.g. AutoPGD-L2). Besides, we apply attacks with different loss functions such as cross entropy (AutoPGD) and difference of logits ratio (AutoPGD-DLR), to avoid the 'fake' adversarial examples caused by gradient vanishing (Athalye et al., 2018).

To measure the quality of trade-off between standard accuracy (SA) and adversarial accuracy (AA), we define **SA-AA front** as an empirical Pareto front for SA and AA. We draw this front by conducting different $\lambda$ on Vanilla AT and different $\gamma$ on CA-AT.

## 5.2 Experimental Results for Adversarial PEFT

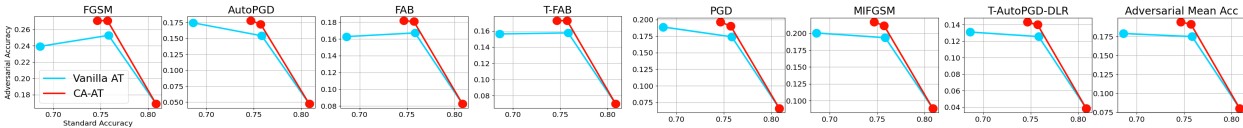

Figure 4: SA-AA Fronts for Adversarial PEFT on Swin-T using Adapter.

Figure 5: SA-AA Fronts for Adversarial PEFT on ViT using Adapter on Stanford Dogs.

**Experimental setup on adversarial PEFT.** For the experiments on adversarial PEFT, we leverage the adversarially pretrained Swin-T and ViT downloaded from ARES[1]. For adapter, we implement it as (Pfeiffer et al., 2020) by inserting an adapter module subsequent to the MLP block at each layer with a reduction factor of 8.

**CA-AT offers the better trade-off on adversarial PEFT.** Fig. 4 shows the SA-AA fronts on fine-tuning robust pretrained Swin Transformer on CUB-Bird and StanfordDogs by using Adapter. We set $\lambda = [0, 0.5, 1]$ for Vanilla AT and $\gamma = [0.8, 0.9, 1]$ for CA-AT. The red data points for CA-AT are positioned in the upper right area relative to the blue points for Vanilla AT. It shows that CA-AT can consistently attain better standard and adversarial accuracy compared to the Vanilla AT across different datasets. Besides, we observed that on fine-grained datasets such as CUB-Bird and Stanford Dogs, the superiority of CA-AT is more significant compared to the results on normal datasets.

**Results for CA-AT with Different Pretrained Models.** Fig. 5 shows that CA-AT can also boost the trade-off performance on ViT. The main difference between these two models is that, ViT treats image patches as tokens and processes them with a standard transformer architecture Vaswani et al. (2017), while Swin-T uses shifted windows for hierarchical feature merging. While ViT applies global attention directly on image patches, Swin Transformer applies local attention within windows and uses a hierarchical approach to better handle larger and more detailed images. The superiority of CA-AT on ViT is not as significant as it is on Swin-T (Fig. 4b), but it still can gain better standard and adversarial accuracy compared to Vanilla AT.

---

[1] https://github.com/thu-ml/ares

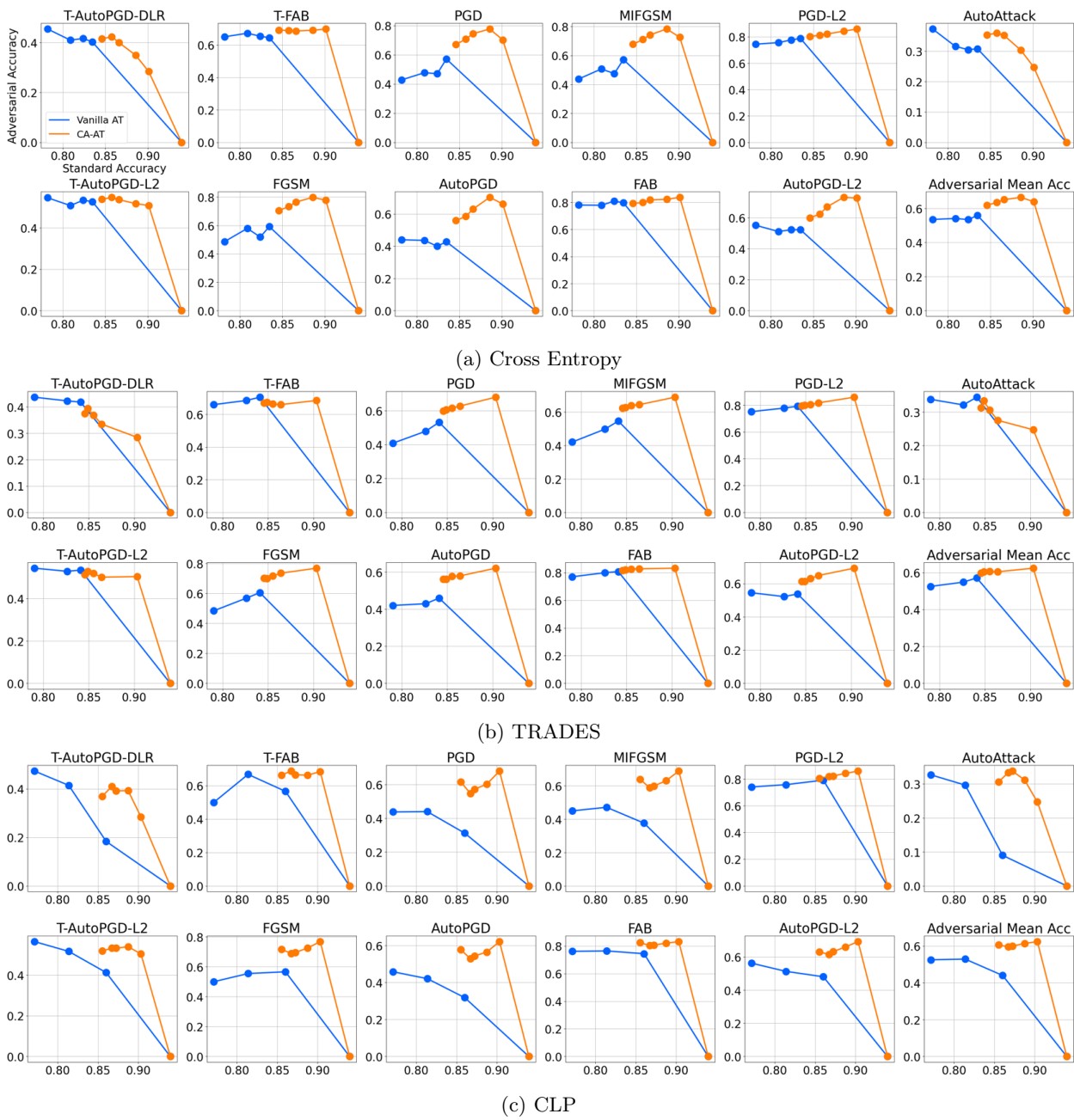

Figure 6: SA-AA Fronts for Adversarial Training from Scratch on CIFAR10 using ResNet18 with Different Adversarial Loss Functions including Cross Entropy, TRADES, and CLP.

## 5.3 Experimental Results for Adversarial Training from Scratch

**CA-AT results in better trade-off with different adversarial loss functions.** Fig. 6a visualizes SA-AA fonts from experiments using vanilla AT with $\lambda = [0, 0.25, 0.5, 0.75, 1]$ and CA-AT with $\gamma = [0.7, 0.75, 0.8, 0.85, 0.9, 1]$ on CIFAR10. In this figure, most orange data points (CA-AT) lie in the upper right space of blue points (Vanilla AT), indicating that CA-AT offers a better empirical Pareto front for the trade-off between standard accuracy and adversarial accuracy. Moreover, Fig. 6c and Fig. 6b show CA-AT can also consistently boost the adversarial accuracy for different adversarial loss functions used in AT such as TRADES (Zhang et al., 2019) and CLP (Kannan et al., 2018). For the experiments on CIFAR100, we

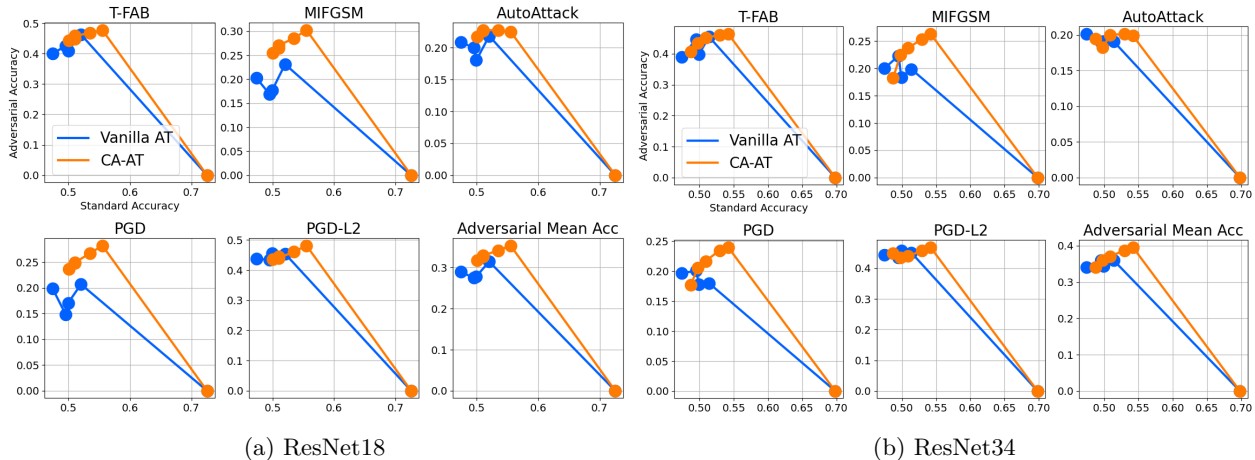

(a) ResNet18           (b) ResNet34

Figure 7: SA-AA Fronts for Adversarial Training from Scratch on CIFAR100.

| | | Standard Accuracy | | PGD | | AutoPGD | | MIFGSM | | FAB | | T-FAB | | FGSM | |
|---|---|---|---|---|---|---|---|---|---|---|---|---|---|---|---|
| | $p = \infty$ | CA-AT | Vanilla AT | CA-AT | Vanilla AT | CA-AT | Vanilla AT | CA-AT | Vanilla AT | CA-AT | Vanilla AT | CA-AT | Vanilla AT | CA-AT | Vanilla AT |
| ResNet18 | 8/255 | | | **0.7442** | 0.4703 | **0.6301** | 0.3996 | **0.7419** | 0.4745 | **0.8177** | 0.809 | **0.6861** | 0.6538 | **0.7649** | 0.519 |
| | 16/255 | **0.8659** | 0.8239 | **0.7311** | 0.4248 | **0.5555** | 0.2486 | **0.7233** | 0.4225 | 0.7475 | **0.78** | **0.5445** | 0.5104 | **0.7435** | 0.4387 |
| | 24/255 | | | **0.7189** | 0.405 | **0.4886** | 0.1963 | **0.7182** | 0.413 | 0.6858 | **0.7333** | **0.4599** | 0.4783 | **0.7235** | 0.403 |
| | 32/255 | | | **0.7033** | 0.3877 | **0.4455** | 0.1589 | **0.7182** | 0.413 | 0.6402 | **0.6836** | **0.4044** | 0.4507 | **0.7066** | 0.379 |
| | | Standard Accuracy | | PGD | | AutoPGD | | MIFGSM | | FAB | | T-FAB | | FGSM | |
| | $p = \infty$ | CA-AT | Vanilla AT | CA-AT | Vanilla AT | CA-AT | Vanilla AT | CA-AT | Vanilla AT | CA-AT | Vanilla AT | CA-AT | Vanilla AT | CA-AT | Vanilla AT |
| ResNet34 | 8/255 | | | **0.8098** | 0.5973 | **0.7285** | 0.4417 | **0.8111** | 0.5983 | **0.8247** | 0.8068 | **0.7274** | 0.6951 | **0.8149** | 0.5327 |
| | 16/255 | **0.8753** | 0.8305 | **0.8034** | 0.5756 | **0.6793** | 0.3395 | **0.8077** | 0.5791 | **0.7738** | 0.7613 | **0.6013** | 0.5762 | **0.7916** | 0.2762 |
| | 24/255 | | | **0.7957** | 0.5602 | **0.6445** | 0.2859 | **0.8067** | 0.5743 | **0.7307** | 0.6937 | **0.5142** | 0.5174 | **0.7743** | 0.1428 |
| | 32/255 | | | **0.785** | 0.5443 | **0.6165** | 0.2498 | **0.8067** | 0.5743 | **0.6918** | 0.6221 | **0.4424** | 0.4822 | **0.7616** | 0.088 |

Table 1: Evaluation results on CIFAR10 for CA-AT ($\gamma = 0.8$) and Vanilla AT ($\lambda = 0.5$) across different $L_\infty$-based attacks with various values of budget $\delta$.

selected the strongest and most representative attack methods to evaluate the model's robustness, including targeted attack (T-FAB), untargeted attacks (PGD, MIFGSM), $L_2$-norm attack (T-PGD), and ensemble attack (AutoAttack). Showing the trade-off results on CIFAR100 in Fig. 7a (ResNet18) and Fig. 7b (ResNet34), the performance gain of CA-AT is more limited compared to the one on CIFAR10, but it can still achieve better performance on standard accuracy and adversarial accuracy against various adversarial attacks.

**CA-AT is more robust to adversarial attacks with larger budget values.** We evaluate adversarial precision through various adversarial attacks with different attack budget values $\delta$, to demonstrate the superiority of our model over Vanilla AT under various intensities of adversarial attacks. We applied both Vanilla AT and CA-AT to ResNet18 on CIFAR10, and the results about $L_\infty$-based attacks are shown in Table 1. In Table 1, although our CA-AT achieves slightly lower adversarial accuracy against FAB when $\delta$ is larger than 8/255, it outperforms the Vanilla AT in both standard accuracy and adversarial accuracy on any other attack methods (e.g. AutoPGD, MIFGSM, and T-FAB) with different budget $\delta$. It clearly illustrates that, compared to Vanilla AT, CA-AT can enhance the model's adversarial robustness ability to resist stronger adversarial attacks with larger budget $\delta$.

**CA-AT enables AT via stronger adversarial examples.** In our toy experiment (Fig. 2), the conflict $\mu$ would be more serious if we utilized adversarial examples with a larger attack budget $\delta$ during AT. It implies that Vanilla AT cannot handle stronger adversarial examples during training because of the gradient conflict. In Fig. 8, we visualize the results of training ResNet34 on CIFAR10 with adversarial samples produced by the same attack method (PGD), but different attack budgets ($\delta = 8/255$ and $\delta = 16/255$), and evaluate the adversarial accuracies against various adversarial methods (e.g. FGSM and PGD) with different budgets (x-axis). Compared to the blue and orange curves (Vanilla AT with $\delta = 8/255$), it shows that Vanilla AT fails when training with the adversarial attack with a higher perturbation bound, causing a decrease in both standard and adversarial accuracy. On the contrary, CA-AT, shown as the green and red curves, can

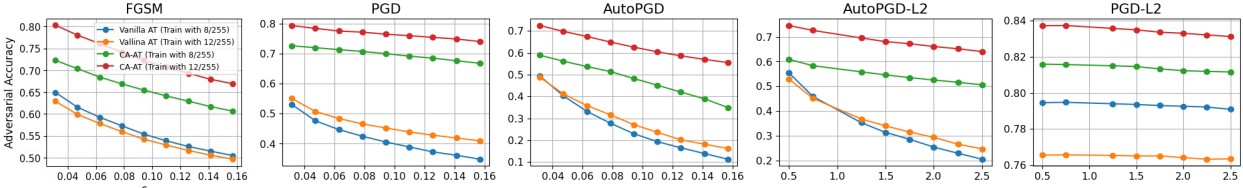

Figure 8: Results for Vanilla AT and CA-AT trained on adversarial samples with two different budget values ($\delta = 8/255, \delta = 12/255$) on CIFAR10 with ResNet18. We evaluate the adversarial accuracy among different adversarial attacks with different budget values $\delta$ denoting as the x-axis.

improve both standard and adversarial accuracy by involving stronger adversarial samples with larger attack budgets.

**Experimental Results in Appendix.** More experimental results for CA-AT regarding different model architectures (WRN-28-10), different attack methods utilized for producing adversarial samples during AT, various $L_2$-based attacks with different budgets, and black-box attacks can be found in Appendix C. In addition, the detailed proof for Theorem 1 is included in Appendix A.

## 6 Conclusion & Outlook

In this work, we illustrate that the weighted-average method in AT is not capable of achieving the 'near-optimal' trade-off between standard and adversarial accuracy due to the gradient conflict existing in the training process. We demonstrate the existence of such a gradient conflict and its relation to the attack budget of adversarial samples used in AT practically and theoretically. Based on this phenomenon, we propose a new trade-off framework for AT called Conflict-Aware Adversarial Training (CA-AT) to alleviate the conflict by gradient operation. Extensive results demonstrate the effectiveness of CA-AT for gaining trade-off results under the setting of training from scratch and PEFT. Considering the cost for gradient operation, CA-AT is more appropriate for adversarial PEFT than full fine-tuning when dealing with very large models like ViT.

For future work, we plan to undertake a more detailed exploration of the gradient conflict phenomenon in AT from the data-centric perspective. We hold the assumption that some training samples can cause serious gradient conflict, while others do not. We will evaluate this assumption in future work and intend to reveal the influence of training samples causing gradient conflict.

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

## A  Proof of Theorem 1

Considering $||\epsilon||_2$ or $||\epsilon||_\infty$ is usually very small for adversarial examples, we utilize Taylor Expansion for $x$ as the approximation for the adversarial loss $\mathcal{L}(x + \epsilon; \theta)$, such that:

$$\mathcal{L}(x + \epsilon; \theta) = \mathcal{L}(x; \theta) + (\frac{\partial \mathcal{L}(x;\theta)}{\partial x})^T \epsilon + \mathcal{O}(||\epsilon||^2) \tag{5}$$

To derive an upper bound on the gradient conflict in the regime that $||\epsilon||$ gets small, we will only consider the first-order term above. We then take the derivative of both sides of the equation with respect to $\theta$ to obtain:

$$g_{\mathrm{a}} = g_{\mathrm{c}} + \frac{\partial^2 \mathcal{L}(x;\theta)}{\partial x \partial \theta} \epsilon = g_{\mathrm{c}} + \frac{\partial g_{\mathrm{c}}}{\partial x} \epsilon = g_{\mathrm{c}} + H\epsilon \tag{6}$$

where $H = \frac{\partial g_{\mathrm{c}}}{\partial x} \in \mathbb{R}^{d_\theta \times d_x}$. $d_\theta / d_x$ denotes the dimension of parameter $\theta$ and input data $x$. By multiplying $g_{\mathrm{a}}^T$ and $g_{\mathrm{c}}^T$ on the two sides of Eq. (6), respectively, we can obtain Eq. (7) and Eq. (8) as follows.

$$g_{\mathrm{c}}^T g_{\mathrm{a}} = ||g_{\mathrm{c}}||_2^2 + g_{\mathrm{c}}^T H \epsilon \tag{7}$$

$$||g_{\mathrm{a}}||_2^2 = g_{\mathrm{a}}^T g_{\mathrm{c}} + g_{\mathrm{a}}^T H \epsilon \tag{8}$$

Eq. (7) minus Eq. (8):

$$g_{\mathrm{c}}^T g_{\mathrm{a}} = \frac{||g_{\mathrm{a}}||_2^2 + ||g_{\mathrm{c}}||_2^2 + \epsilon^T H^T (g_{\mathrm{c}} - g_{\mathrm{a}})}{2} \tag{9}$$

Based on Eq. (6), we can replace $(g_{\mathrm{c}} - g_{\mathrm{a}})$ as $H\epsilon$:

$$g_{\mathrm{c}}^T g_{\mathrm{a}} = \frac{||g_{\mathrm{a}}||_2^2 + ||g_{\mathrm{c}}||_2^2 - \epsilon^T H^T H \epsilon}{2} \tag{10}$$

Recall the definition of $\mu$ as $\mu = ||g_{\mathrm{c}}||_2 \cdot ||g_{\mathrm{a}}||_2 \cdot (1 - \cos(g_{\mathrm{c}}, g_{\mathrm{a}}))$

$$
\begin{aligned}
\mu &= ||g_{\mathrm{c}}||_2 \cdot ||g_{\mathrm{a}}||_2 \cdot (1 - \cos(g_{\mathrm{c}}, g_{\mathrm{a}})) \\
&= ||g_{\mathrm{c}}||_2 \cdot ||g_{\mathrm{a}}||_2 - g_{\mathrm{c}}^T g_{\mathrm{a}} \\
&= \frac{2||g_{\mathrm{c}}||_2 \cdot ||g_{\mathrm{a}}||_2 - ||g_{\mathrm{a}}||_2^2 - ||g_{\mathrm{c}}||_2^2 + \epsilon^T H^T H \epsilon}{2} \quad \text{(Use Eq. (10))} \\
&= \frac{\epsilon^T \mathcal{K}(\theta, x)\epsilon - (||g_{\mathrm{c}}||_2 - ||g_{\mathrm{a}}||_2)^2}{2} \leq \frac{\epsilon^T \mathcal{K}(\theta, x)\epsilon}{2} \leq \frac{\lambda_{max} \epsilon^T \epsilon}{2}
\end{aligned} \tag{11}
$$

where $\mathcal{K}(\theta, x) = H^T H$ is a symmetric and positive semi-definite matrix, and $\lambda_{max}$ is the largest eigenvalue of $K$, where $\lambda_{max} \geq 0$.

Considering two widely-used restrictions for perturbation $\epsilon$ applied in adversarial examples as $l_2$ and $l_\infty$ norm, we have:

- For $||\epsilon||_2 \leq \delta$, where $\mu \leq \frac{1}{2}\lambda_{max}\delta^2$. The upper bound of $\mu$ is $\mathcal{O}(\delta^2)$.

- For $||\epsilon||_\infty \leq \delta$, it implies that the absolute value of each element of $\epsilon$ is bounded by $\delta$, where $\epsilon^T \epsilon = \sum_{i=0}^d \epsilon_i^2 \leq d^2\delta^2$. The upper bound of $\mu$ is $\mathcal{O}(d^2\delta^2)$.

## B  Analytical Solution for the Inner Maximization

We introduce the details about how to get the analytical inner-max solution (Eq. (3)) for our synthetic experiment presented in Section 3. As we introduced in Section 3, consider a linear model as $f(x) = w^T x + b$

| | Standard | T-AutoPGD-DLR | T-AutoPGD-L2 | T-FAB | FGSM | PGD | AutoPGD | MIFGSM | FAB | PGD-L2 | AutoPGD-L2 | AutoAttack | Adversarial Mean Acc |
|---|---|---|---|---|---|---|---|---|---|---|---|---|---|
| $\gamma = 0.8$, PGD | **0.8659** | 0.4004 | 0.5356 | 0.6861 | **0.7649** | **0.7442** | 0.6301 | **0.7419** | **0.8177** | **0.8211** | 0.67 | **0.3517** | **0.6512** |
| $\gamma = 0.8$, PGD-DLR | 0.8646 | **0.4147** | **0.539** | **0.7168** | 0.7452 | 0.672 | **0.6429** | 0.6864 | 0.8001 | 0.8196 | **0.6919** | 0.3260 | 0.6413 |
| $\gamma = 0.9$, PGD | **0.9009** | 0.2844 | 0.5075 | **0.6986** | 0.7781 | 0.7021 | **0.6624** | 0.7251 | **0.8371** | **0.8588** | 0.7267 | 0.2472 | 0.6389 |
| $\gamma = 0.9$, PGD-DLR | 0.8923 | **0.3794** | **0.5353** | 0.6874 | **0.779** | **0.7207** | 0.6428 | **0.7315** | 0.8229 | 0.8488 | 0.7038 | **0.2992** | **0.6501** |

Table 2: Evaluation results for CA-AT for using different inner maximization solver (PGD/PGD-DLR) during the process of AT.

under a binary classification task where $y \in \{+1, -1\}$. The predicted probability of sample $x$ with respect to its ground truth $y$ can be defined as:

$$p(y|x) = \frac{1}{1 + \exp(-y \cdot f(x))} \tag{12}$$

Then, the BCE loss function for sample $x$ can be formulated as:

$$\mathcal{L}(f(x), y) = -\log(p(y|x)) = \log(1 + \exp(-y \cdot f(x))) \tag{13}$$

Consider the perturbation $\epsilon$ under the restriction of $L_\infty$ norm, the adversarial attack for such a linear model can be formulated as an inner maximization problem as Eq. (14).

$$\max_{\|\epsilon\|_\infty \leq \delta} \log(1 + \exp(-y \cdot f(x + \epsilon))) \equiv \min_{\|\epsilon\|_\infty \leq \delta} y \cdot w^T \epsilon \tag{14}$$

Consider the case that $y = +1$, where the $L_\infty$ norm says that each element in $\epsilon$ must have magnitude less than or equal $\delta$, we clearly minimize this quantity when we set $\epsilon_i = -\delta$ for $w_i \geq 0$ and $\epsilon_i = \delta$ for $w_i < 0$. For $y = -1$, we would just flip these quantities. That is, the optimal solution $\epsilon^*$ to the above optimization problem for the $L_\infty$ norm is expressed as Eq. (15).

$$\epsilon^* = -y \cdot \delta \odot \text{sign}(w) \tag{15}$$

where $\odot$ is the element-wise multiplication. Based on Eq. (15), we can formulate the adversarial loss as follows, which is as same as the adversarial loss presented in Eq. (3).

$$\begin{aligned}
\mathcal{L}(f(x + \epsilon^*), y) &= \log(1 + \exp(-y \cdot w^T x - y \cdot b - y \cdot w^T \epsilon^*)) \\
&= \log(1 + \exp(-y \cdot f(x) + \delta \|w\|_1)) \tag{16}
\end{aligned}$$

## C  Additional Experimental Results

**The effect of inner maximization solver in AT.** In Table 2, we conduct the ablation study for using different attack methods to generate adversarial samples during adversarial training from scratch. We find that PGD-DLR can achieve higher adversarial accuracies when $\gamma = 0.9$ but lead them worse when $\gamma = 0.8$ but not significant. We conclude that the effect of the inner maximization solver, as well as the adversarial attack method during AT, does not dominate the performance of CA-AT.

**Results for different model architectures.** For different model architectures such as ResNet34 and WRN-28-10, their SA-AA front on CIFAR10 and and CIFAR100 with different adversarial loss functions are shown in Fig. 9, Fig. 11, and Fig. 10. All of those figures demonstrate CA-AT can consistently surpass Vanilla AT across different model architectures.

**Results for $L_2$-based adversarial attacks with different budgets.** Besides evaluating the adversarial accuray on $L_\infty$-based attacks with different budgets (Table 1), we also evaluate the adversarial robustness against $L_2$-based adversarial attacks with different budgets ($\|\epsilon\|_2 = [0.5, 1, 1.5, 2]$), which is shown in Table 4.

**Results for Black-box Attack & Optimization-based Attack**. To further evaluate the robustness of CA-AT against optimization-based attacks, and also demonstrate that the performance gain of adversarial accuracy is not brought by obfuscated gradients (Athalye et al., 2018), we evaluate the adversarial robustness

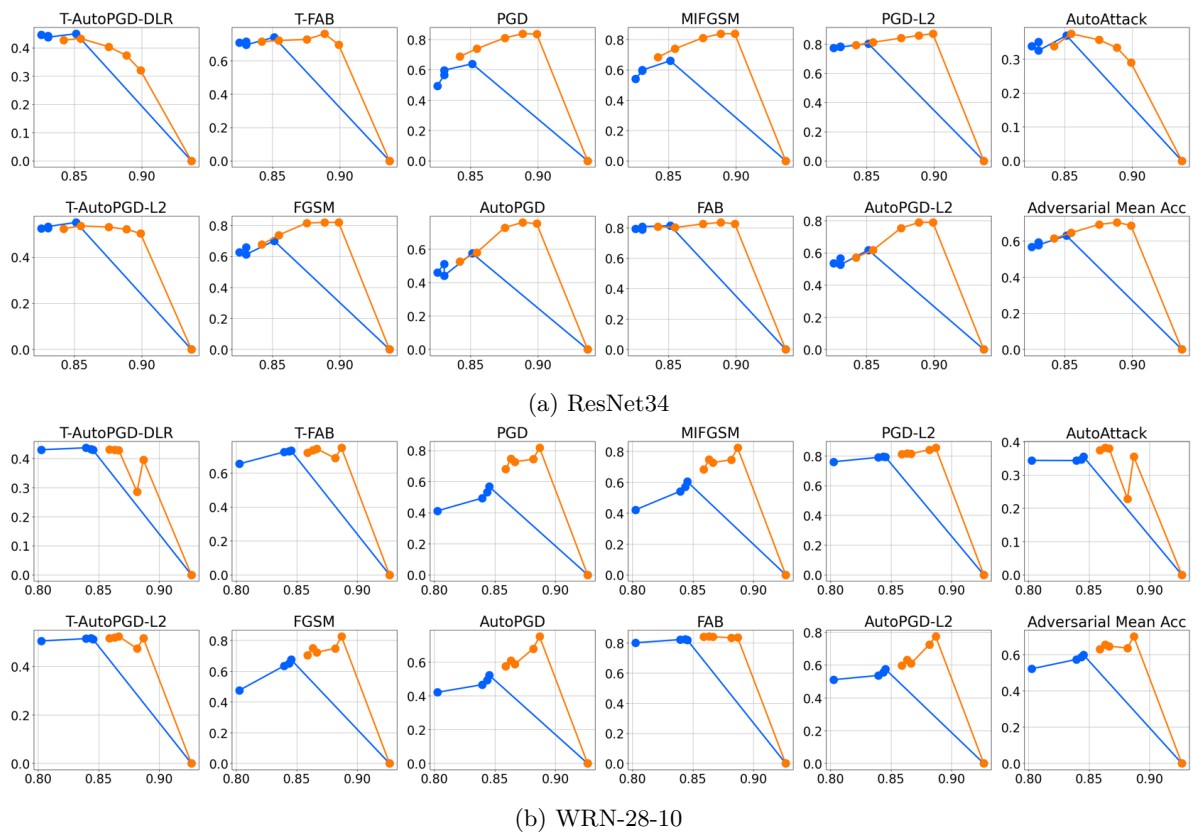

Figure 9: SA-AA Fronts for Adversarial Training from Scratch on CIFAR10 with ResNet34 and WRN-28-10.

|  |  | Standard Accuracy | DDN Attack | C&W Attack | Square Attack |
|---|---|---|---|---|---|
| ResNet18 | Standard Training | 0.9392 | 0.133 | 0.1171 | 0.6795 |
|  | Vanilla AT, $\lambda = 0.5$ | 0.8239 | 0.4585 | 0.4565 | 0.7329 |
|  | CA-AT, $\gamma = 0.8$ | **0.8659** | **0.5991** | **0.5089** | **0.7656** |
| ResNet 34 | Standard Training | 0.9363 | 0.1036 | 0.088 | 0.6771 |
|  | Vanilla AT, $\lambda = 0.5$ | 0.8305 | 0.5411 | 0.4747 | 0.7429 |
|  | CA-AT, $\gamma = 0.8$ | **0.8753** | **0.7207** | **0.4885** | **0.7934** |

Table 3: The comparison Results against Black-box Attack (Square Attack) and Optimization-based Attack (C&W Attack and DDN Attack) between Vanilla AT and CA-AT on CIFAR10. For standard training, the results for standard performance were marked as blue.

via black-box attack (Square Andriushchenko et al. (2020)) and optimization-based attack (C&W Carlini & Wagner (2017), DDN Rony et al. (2019)). Table 3 shows that CA-AT ($\gamma = 0.8$) outperforms Vanilla AT ($\lambda = 0.5$) on defending against the both black-box attack and optimization-based attack, while achieving higher standard accuracy.

**More Advanced Adversarial Loss.**,Besides TRADES and CLP, we conducted more experiments on MART Wang et al. (2019) shown in Table 5.

**Ablation Study on Learning Rate and Batch Size**. We conducted the ablation study on different training parameters such as learning rate and batch size in Fig. 12. The observation is that although batch size and learning rate affect the standard accuracy and adversarial accuracies against various attacks, CA-AT can consistently lead to better standard performance and adversarial robustness across different batch sizes and learning rates.

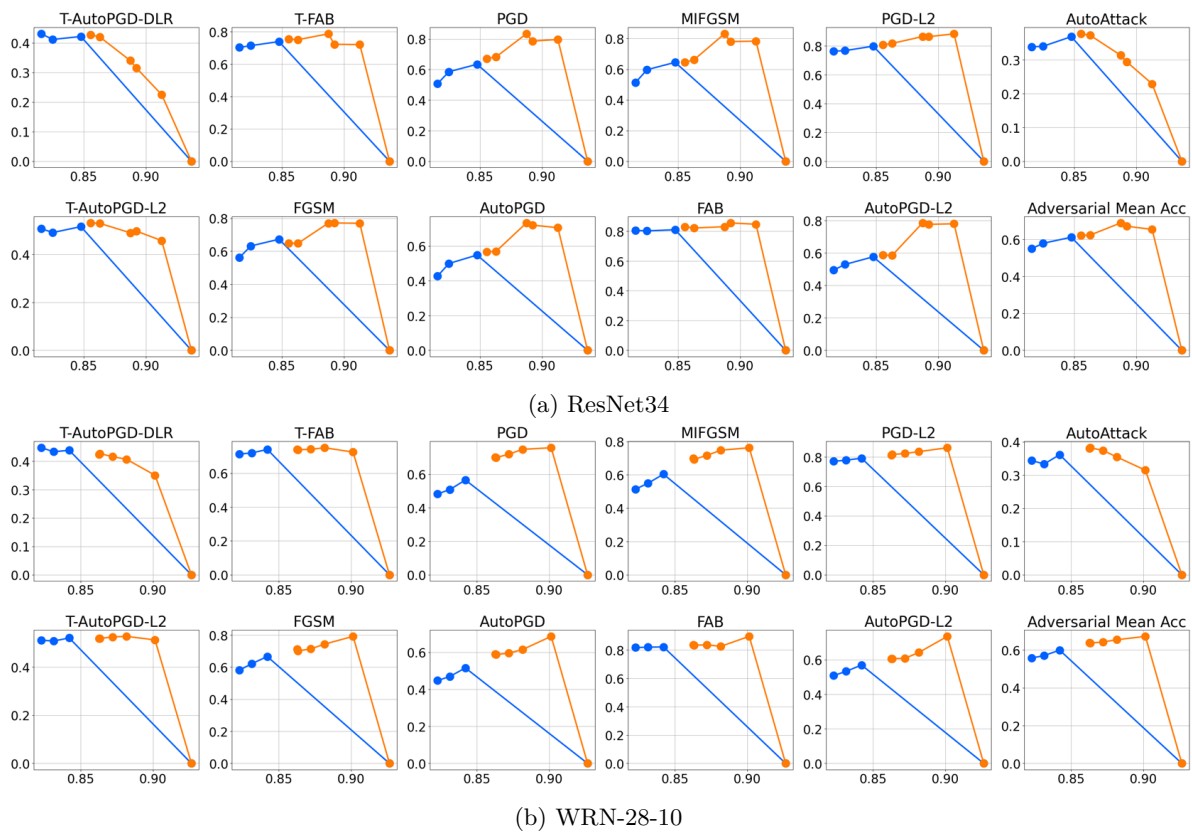

Figure 10: SA-AA Fronts on CIFAR10 for Adversarial Training from Scratch using TRADES with ResNet34 and WRN-28-10.

|  |  | PGD-L2 | | AutoPGD-L2 | | T-AutoPGD-L2 | |
|---|---|---|---|---|---|---|---|
|  | $p = 2$ | $\gamma = 0.8$ | $\lambda = 0.5$ | $\gamma = 0.8$ | $\lambda = 0.5$ | $\gamma = 0.8$ | $\lambda = 0.5$ |
| ResNet18 | 0.5 | **0.8211** | 0.7759 | **0.67** | 0.5222 | **0.5356** | 0.5327 |
|  | 1 | **0.8207** | 0.7748 | **0.603** | 0.3036 | 0.261 | **0.2762** |
|  | 1.5 | **0.8194** | 0.7738 | **0.5652** | 0.2405 | **0.1483** | 0.1428 |
|  | 2 | **0.8187** | 0.7734 | **0.5331** | 0.2115 | **0.0904** | 0.088 |
|  |  | PGD-L2 | | AutoPGD-L2 | | T-AutoPGD-L2 | |
|  | $p = 2$ | $\gamma = 0.8$ | $\lambda = 0.5$ | $\gamma = 0.8$ | $\lambda = 0.5$ | $\gamma = 0.8$ | $\lambda = 0.5$ |
| ResNet34 | 0.5 | **0.8411** | 0.78 | **0.7534** | 0.5255 | 0.5301 | **0.5325** |
|  | 1 | **0.8412** | 0.7791 | **0.7249** | 0.3806 | 0.2571 | **0.2683** |
|  | 1.5 | **0.8403** | 0.7784 | **0.7022** | 0.3462 | **0.1446** | 0.1438 |
|  | 2 | **0.8386** | 0.7781 | **0.679** | 0.3196 | 0.0899 | **0.0905** |

Table 4: Evaluation Results for CA-AT ($\gamma = 0.8$) and vanilla AT ($\lambda = 0.5$) across different $L_2$-based attacks with various restriction $\theta$.

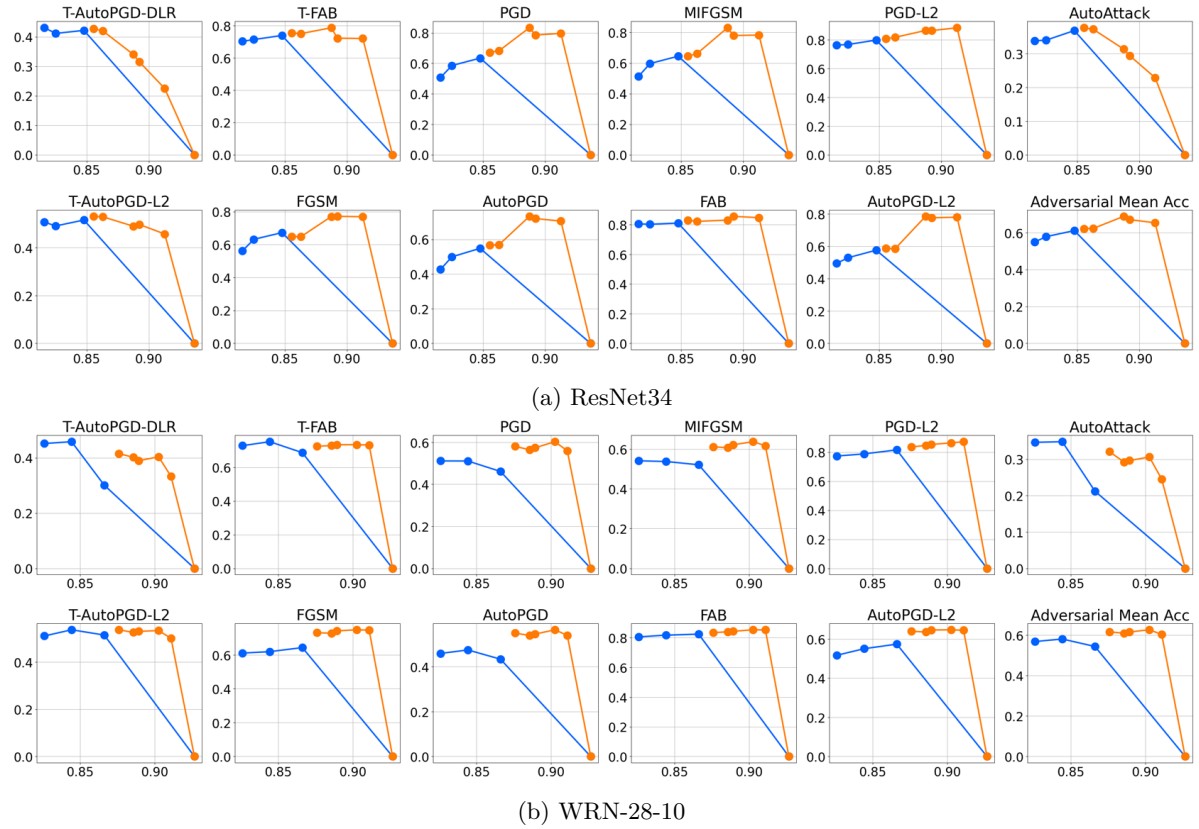

Figure 11: SA-AA Fronts for Adversarial Training from Scratch on CIFAR10 using CLP with ResNet34 and WRN-28-10.

| | | Standard | T-AutoPGD | T-FAB | FGSM | PGD | AutoPGD | MIFGSM | FAB |
|---|---|---|---|---|---|---|---|---|---|
| ResNet 18 | Vanilla AT, $\lambda = 0.5$ | 0.83 | 0.5344 | 0.657 | 0.6017 | 0.5343 | 0.2666 | 0.4483 | 0.55 |
| | CA-AT, $\gamma = 0.8$ | **0.8848** | **0.5381** | **0.6826** | **0.7953** | **0.782** | **0.669** | **0.7859** | **0.8151** |
| ResNet 34 | Vanilla AT, $\lambda = 0.5$ | 0.82 | 0.3624 | 0.503 | 0.4832 | 0.3466 | 0.2479 | 0.3359 | 0.6284 |
| | CA-AT, $\gamma = 0.8$ | **0.8857** | **0.4922** | **0.7357** | **0.8297** | **0.8466** | **0.7687** | **0.8466** | **0.8299** |

Table 5: Results for Training with MART loss on CIFAR10 with ResNet18

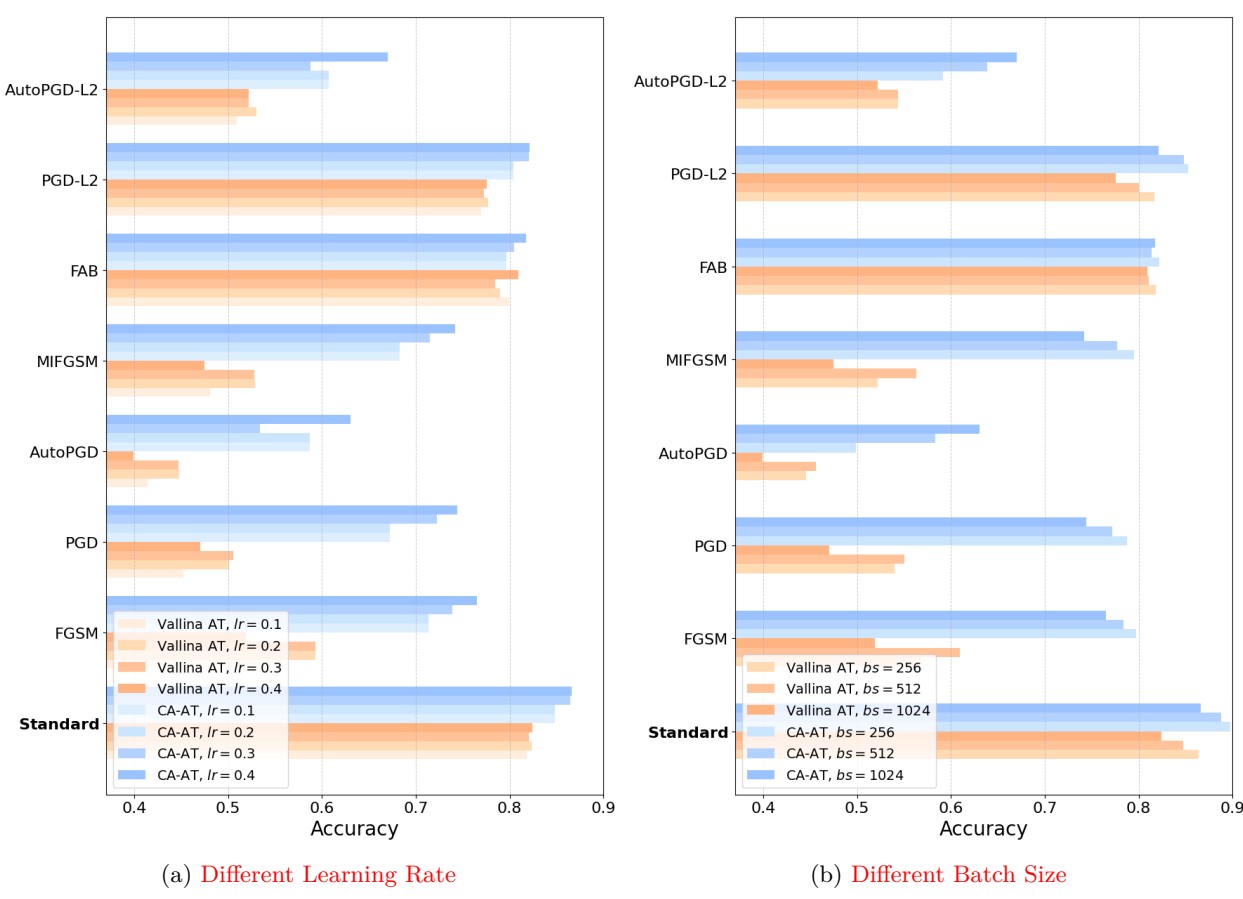

(a) Different Learning Rate

(b) Different Batch Size

Figure 12: Ablation Study for Different Training Hyperparameters including Learning Rate and Batch Size.

