# OpenReview forum: "Conflict-Aware Adversarial Training"
_TMLR — Rejected by TMLR_

### Review · Reviewer_5n4Z · 2025-02-19

**Summary Of Contributions:**

This paper presents an approach to adversarial training called Conflict-Aware Adversarial Training (CA-AT). Traditional adversarial training methods, which use a weighted-average approach to balance standard and adversarial accuracy, are suboptimal due to gradient conflicts. The proposed method aims to mitigate this conflict by introducing a trade-off factor based on the angle between standard and adversarial gradients. The theoretical analysis and experiments show that CA-AT improves the trade-off between standard and adversarial accuracy in both training from scratch and parameter-efficient finetuning settings.

**Audience:**

Yes

**Broader Impact Concerns:**

The paper does not include a broader impact discussion. While the proposed method enhances adversarial robustness, it could potentially be used to develop more resilient adversarial attacks. Additionally, adversarial training has implications for security-sensitive applications, where unintended consequences could be significant. The authors should address these concerns explicitly.

**Claims And Evidence:**

No

**Requested Changes:**

1.	Provide additional empirical validation of the theoretical analysis by demonstrating its applicability on real-world datasets beyond synthetic experiments. Specifically, show how gradient conflict manifests in large-scale datasets and whether CA-AT successfully mitigates it in practice.
2.	Include a more detailed ablation study on the choice of the trade-off factor γ. Further analyses would clarify whether the method is robust to different values of this hyperparameter.
3.	Discuss the computational overhead of CA-AT in greater detail. Provide training time comparisons and resource consumption metrics relative to standard adversarial training methods.
4.	Compare CA-AT against recent adaptive adversarial training methods that dynamically adjust trade-off factors during training. This would help establish whether CA-AT offers meaningful advantages over existing state-of-the-art approaches.
5.	Improve the presentation of figures and tables. Ensure that all experimental results are clearly labeled and include additional explanations where necessary to aid comprehension.
6.	Add a broader impact statement discussing ethical concerns, potential risks, and any unintended consequences of the proposed method. This is crucial given the growing concerns around adversarial robustness and security.

**Strengths And Weaknesses:**

Strengths:

1.	The paper introduces a well-motivated method to address gradient conflicts in adversarial training.
2.	The authors provide a solid theoretical foundation, including a formal definition of gradient conflict and its relation to the attack budget.
3.	The proposed method demonstrates consistent improvements in both standard and adversarial accuracy across different settings.

Weaknesses:

1.	The analysis, while insightful, lacks a rigorous empirical validation beyond synthetic datasets, making it difficult to assess its generalizability to real-world settings.
2.	The impact of different hyperparameters, particularly the choice of the trade-off factor, is not thoroughly explored. It is unclear how sensitive the method is to this factor.
3.	The paper does not adequately discuss the computational cost of CA-AT compared to standard adversarial training. Although it claims no additional training cost, more detailed profiling of time and resource consumption is needed.
4.	Some experimental comparisons lack proper baselines. For example, the paper does not compare against more recent adaptive adversarial training strategies that dynamically adjust the trade-off factor.
5.	The figures and tables presenting the experimental results are difficult to interpret. A more structured presentation with clearer explanations would improve readability.
6.	The broader impact discussion is missing, leaving ethical implications and potential misuse of the proposed method unaddressed.

---

> ### Author Response · Authors · 2025-03-17
>
> **Comment:** The analysis, while insightful, lacks rigorous empirical validation beyond synthetic datasets, making it difficult to assess its generalizability to real-world settings. Provide additional empirical validation of the theoretical analysis by demonstrating its applicability on real-world datasets beyond synthetic experiments. Specifically, show how gradient conflict manifests in large-scale datasets and whether CA-AT successfully mitigates it in practice.
>
> **Response:** Please refer to Figure 3 for empirical validation. In this Figure, we demonstrated that (1) the gradient conflict does exist during adversarial training, and (2) our method (CA-AT) can mitigate gradient conflict.
> Besides, we would like to highlight the reason why we conducted the analysis on the toy example.
>
> (1) Under the scenario of logistic regression models, we can have an analytical solution for the inner maximization, which can de-factorize the influence of solving inner maximization. (Eq 3)
>
> (2) Visualization of low-dimensional gradient can reflect the geometric information in a more promising way. (Figure 2(a))

---

> ### Author Response · Authors · 2025-03-17
>
> **Comment:** The impact of different hyperparameters, particularly the choice of the trade-off factor, is not thoroughly explored. It is unclear how sensitive the method is to this factor. Include a more detailed ablation study on the choice of the trade-off factor. Further analyses would clarify whether the method is robust to different values of this hyperparameter.
>
> **Response:** Thank you for raising this point. We have experimental results for different trade-off factors (Figure 4,5,6,7). We drew the Pareto Front for different trade-off factors showing that CA-AT consistently improves both standard performance and adversarial robustness.

---

> ### Author Response · Authors · 2025-03-17
>
> **Comment:** The paper does not adequately discuss the computational cost of CA-AT compared to standard adversarial training. Although it claims no additional training cost, more detailed profiling of time and resource consumption is needed. Discuss the computational overhead of CA-AT in greater detail. Provide training time comparisons and resource consumption metrics relative to standard adversarial training methods.
>
> **Response:** We acknowledge the importance of quantifying computational efficiency. Regarding time complexity, CA-AT modifies only the gradient update step and does not require additional forward or backward passes, leading to negligible computational overhead. Similarly, the space complexity remains unchanged. Below, we provide empirical comparisons conducted on an RTX A6000 GPU.
>
> |                           | Averaged Time Cost for Each Epoch | Memory (Batch Size=1204, ResNet18) |
> |---------------------------|:---------------------------------:|:----------------------------------:|
> | Vanilla AT, $\lambda=0.5$ |           276.62 seconds          |               13513MB              |
> |    CA-AT, $\gamma=0.8$    |           277.31 seconds          |               13513MB              |

---

> ### Author Response · Authors · 2025-03-17
>
> **Comment:** Some experimental comparisons lack proper baselines. For example, the paper does not compare against more recent adaptive adversarial training strategies that dynamically adjust the trade-off factor. Compare CA-AT against recent adaptive adversarial training methods that dynamically adjust trade-off factors during training. This would help establish whether CA-AT offers meaningful advantages over existing state-of-the-art approaches.
>
> **Response:** Thank you for this suggestion. We would like to clarify that “Vanilla AT” refers to a family of adversarial training methods (e.g., Cross Entropy, TRADES, and CLP) based on a linearly hybrid loss (Eq. (2)). In addition, we have now included results for MART [1] in Table 5 of the Appendix. These new results further demonstrate the superiority of CA-AT.
>
> For the methods that dynamically adjust the trade-off factor in adversarial training, we believe our work is the first one to try this according to our literature review. Please let us know if you find any related work that dynamically adjusts the trade-off factor during adversarial training. Thank you.

---

> ### Author Response · Authors · 2025-03-17
>
> **Comment:** The figures and tables presenting the experimental results are difficult to interpret. A more structured presentation with clearer explanations would improve readability. Improve the presentation of figures and tables. Ensure that all experimental results are clearly labeled and include additional explanations where necessary to aid comprehension.
>
> **Response:** Thank you for the feedback. We have improved the tables and figures. While we will improve the captions, axis labels, and formatting for better readability, we would greatly appreciate further clarification on which specific figures or tables you found difficult to interpret. This would help us make targeted improvements to enhance clarity and comprehension.

---

> ### Author Response · Authors · 2025-03-17
>
> **Comment:** The broader impact discussion is missing, leaving ethical implications and potential misuse of the proposed method unaddressed. Add a broader impact statement discussing ethical concerns, potential risks, and any unintended consequences of the proposed method. This is crucial given the growing concerns around adversarial robustness and security.
>
> **Response:** We acknowledge the importance of discussing the broader impact. CA-AT aims to improve adversarial robustness for image classification, which has implications in security-sensitive applications such as medical image analysis systems for healthcare. The discussion of the border impact is shown as follows.
>
> We would like to clarify that CA-AT is fundamentally a defense method designed to improve adversarial robustness, rather than an attack method. As such, the direct risk of harm from CA-AT itself is minimal, since its primary goal is to enhance the security of machine learning systems against adversarial threats. However, there exists the potential for malicious attackers to exploit the insights gained from CA-AT to design stronger adversarial attacks. To mitigate such risks, we encourage the responsible use of CA-AT in security-sensitive applications.

---

### Review · Reviewer_nEQo · 2025-03-03

**Summary Of Contributions:**

This paper investigates the observed gradient conflict in adversarial training, which leads to degraded adversarial performance in real-world applications. The finding that the conflicting gradients of the standard loss and the adversarial loss contribute to this degraded performance leads to the development of a new trade-off factor aimed at improving adversarial training, called “CA-AT” (Conflict-Aware Adversarial Training). This trade-off factor is based on the cosine similarity between the clean and adversarial gradients. If the cosine similarity is greater than a predefined threshold, the adversarial gradients are projected onto a cone defined by the clean gradients and optimized jointly with  the clean gradients. Otherwise, only the clean gradients are optimized. This paper evaluates the proposed gradients updating mechanism on datasets such as CUB-Bird, Stanford Dogs, and CIFAR-10, CIFAR-100, using different neural network architectures (CNNs and pretrained Transformer) with various adversarial budgets in the adversarial training.

**Audience:**

Yes

**Claims And Evidence:**

Yes

**Requested Changes:**

1. Review the text to improve the  grammar and address clarity points to improve the paper's readability.
2. Provide an explanation for the case when $\gamma = 1$ and $\lambda = 1$.
3. Incorporate plots that illustrate the behavior of the gradient over training time
4. Include information about the training time.
5. It would be very valuable if the additional experiments would be included.

**Strengths And Weaknesses:**

**Strengths**:
- This paper addresses an important topic in adversarial robustness by improving the gradient update in adversarial training.
- The proposed gradient update demonstrates improved robustness, particularly in cases with increased perturbation budgets.
- Extensive experiments were conducted which show the effectiveness of the proposed method.

**Weaknesses**:
- In the case of $\gamma = 1$, $g_*$ always equals $g_c$. Additionally, for $\lambda = 1$, the loss focuses exclusively on the adversarial loss. So why  is the adversarial accuracy the same in both cases (see e.g. Fig. 4)?
- I would appreciate a discussion on robust overfitting [1] and how the proposed gradient update can mitigate that.
- How does the training time here compare to that of vanilla adversarial training (AT)?

- It would be interesting to see plots demonstrating the gradient behavior over training time on a single example.

- Conducting experiments on ImageNet would be valuable, as its frequency distribution differs significantly from that of CIFAR-10 [2].

*Clarity points*:
- Some sections contain similar content, leading to redundancy (e.g. page 4  the last two paragraphs)
- The figure captions are often either too lengthy, making it challenging to grasp the key idea (see Figs 1-3), or too brief (see Figs. 4-7). The brevity complicates the understanding of the figures without referring back to the text. Furthermore, the legends in the latter figures are difficult to read, and the values to the markers should be clarified.


[1] Leslie Rice, Eric Wong, and Zico Kolter. Overfitting in adversarially robust deep learning. Proceedings of the 37th International Conference on Machine Learning, 2020

[2] Shishira R Maiya, Max Ehrlich, Vatsal Agarwal, Ser-Nam Lim, Tom Goldstein and Abhinav Shrivastava. Unifying the Harmonic Analysis of Adversarial Attacks and Robustness. 34th British Machine Vision Conference 2023

---

> ### Author Response · Authors · 2025-03-17
>
> **Comment:** In the case $\gamma = 1$, $g_{*}$ always equals $g_{c}$. Additionally, for $\lambda=1$, the loss focuses exclusively on the adversarial loss. So why is the adversarial accuracy the same in both cases (see e.g. Fig. 4)? Provide an explanation for the case when $\gamma = 1$ and $\lambda=1$.
>
> **Response:** We would like to clarify that the overlapped points in Pareto Fronts you see in Figure 4/6 represent the standard training, where $\lambda=0$ for Vanilla AT and $\gamma=1$ for CA-AT. Moreover, for all the overlapped points, no adversarial examples are involved in training, which explains why the adversarial accuracies are close to zero. Thank you for pointing this out. We added a few sentences in our draft to clarify this based on your comments.

---

> ### Author Response · Authors · 2025-03-17
>
> **Comment:** I would appreciate a discussion on robust overfitting and how the proposed gradient update can mitigate that.
>
> **Response:** Thank you for the insightful suggestion. Please refer to Figure 13 in the Appendix, where we add results on robust overfitting. In summary, CA-AT alleviates overfitting on both clean and adversarial examples by dynamically adjusting the gradient updates based on conflict-aware factors, preventing excessive adaptation to adversarial perturbations.

---

> ### Author Response · Authors · 2025-03-17
>
> **Comment:** Review the text to improve the grammar and address clarity points to improve the paper's readability.
>
> **Response:** We appreciate this suggestion. We have carefully proofread the manuscript to improve readability and corrected grammatical mistakes.

---

> ### Author Response · Authors · 2025-03-17
>
> **Comment:** How does the training time here compare to that of vanilla adversarial training (AT)?
> **Response:** The training time for CA-AT is nearly identical to that of Vanilla AT. Below, we provide a comparison evaluated on a single RTX A6000 GPU. You can also run our code in the the anonymous github repo (https://anonymous.4open.science/r/CA-AT-Light-TMLR) to evalute it.
>
> |                           | Averaged Time Cost for Each Epoch | Memory (Batch Size=1204, ResNet18) |
> |---------------------------|:---------------------------------:|:----------------------------------:|
> | Vanilla AT, $\lambda=0.5$ |           276.62 seconds          |               13513MB              |
> |    CA-AT, $\gamma=0.8$    |           277.31 seconds          |               13513MB              |
>
> As shown, CA-AT introduces negligible computational overhead.

---

> ### Author Response · Authors · 2025-03-17
>
> **Comment:** It would be interesting to see plots demonstrating the gradient behavior over training time on a single example.
>
> **Response:** Please refer to Figure 2 for the gradient behavior over the single example on binary classification on logistic regression.
>
> **Comment:** Conducting experiments on ImageNet would be valuable, as its frequency distribution differs significantly from that of CIFAR-10.
>
> **Response:** Thank you for the comments. Please refer to Table 6 for the experimental results on TinyImageNet, which is a subset of ImageNet.

---

> ### Author Response · Authors · 2025-03-17
>
> **Comment:** It would be very valuable if the additional experiments were included.
>
> **Response:** We have added multiple new experiments, which are highlighted in red in the Appendix. These include:
>
> 1.	Additional evaluations using more advanced adversarial loss functions for Vanilla AT.
> 2.	Ablation studies on learning rate and batch size.
> 3.	Experimental results on a larger dataset (Tiny-ImageNet).
> 4.	Discussion on robust overfitting.
> 5.	Evaluation of CA-AT’s robustness against PGD attacks with different step sizes.

---

### Review · Reviewer_5aHv · 2025-03-08

**Summary Of Contributions:**

The authors propose an improved method for adversarial training by trading off the conflict between the gradients between the clean loss objective and adversarial loss objective. They adopt the approach of Yu et al 2020 for gradient surgery used originally for multi-task learning. Empirical evaluations show that the new approach improves both the clean and adversarial accuracies, sometimes by a substantial amount, over plain adversarial training.

**Audience:**

Yes

**Claims And Evidence:**

No

**Requested Changes:**

- Please address the questions above regarding the experiments. If possible, the authors could also consider sharing the code/model to the reviewers to run independent validation of the results.

**Strengths And Weaknesses:**

- The authors recognize that the gradient surgery approached used in Yu et al 2020 for resolving conflicts in multi-task learning can be adopted for resolving conflicts between clean accuracy and adversarial accuracy in adversarial training. This is a very interesting insight and has not been attempted before.

- The empirical results also show promise of the method, with increased clean and adversarial accuracies across different datasets and attack budgets.

- One weakness of the paper is the potential reliability of the empirical results. This is in part due to some of the numbers reported in the tables and charts, and in part due to many previous papers claiming adversarial robustness only found out to be non-robust later. Here are some of my questions for the authors:
    i. In Table 1, why is there such a large gap between PGD and AutoPGD? How many PGD steps do the authors use? There should not be such a large gap if proper step size and number of steps (e.g., those adopted in literature) are used.
    ii. Is AutoPGD the same as the one implemented in AutoAttack? If so why don't the authors just report the numbers from the full suite of AutoAttack methods?
    iii. The adversarial accuracy under AutoPGD seems very high for ResNet18 in Table 1 (for 8/255 on CIFAR10, 63%). Other researchers can only get similar numbers with the use of unlabeled data [1].
[1] Unlabeled Data Improves Adversarial Robustness, Carmon et al 2019
    iv. In Table 1, why is there only one clean accuracy for CA-AT and vanilla AT under different attack budgets? The clean accuracies for these different models should be different.

---

> ### Author Response · Authors · 2025-03-17
>
> We truly appreciate your review, and here are our responses to the proposed questions. Hope our responses can address your concerns.
>
> **Comment:** In Table 1, why is there such a large gap between PGD and AutoPGD? How many PGD steps do the authors use? There should not be such a large gap if proper step size and number of steps (e.g., those adopted in literature) are used.
>
> **Response:** Thank you for your observation. In our experiments, we use a 10-step PGD attack (PGD-10) with a step size of 2/255, following standard settings in the literature [1]. The large gap between PGD and AutoPGD arises because AutoPGD utilizes an adaptive step size and momentum-based optimization, making it significantly stronger than PGD-10. This phenomenon has also been observed in prior works [2]. We also conducted additional experiments using PGD attacks with more steps (e.g., PGD-20/PGD-30/PGD-40) and have updated the results in our draft accordingly. Please refer to Table 8 in the Appendix.
>
> [1] Towards deep learning models resistant to adversarial attacks.
>
> [2] Reliable evaluation of adversarial robustness with an ensemble of diverse parameter-free attacks

---

> ### Author Response · Authors · 2025-03-17
>
> **Comment:** Is AutoPGD the same as the one implemented in AutoAttack? If so why don't the authors just report the numbers from the full suite of AutoAttack methods?
>
> **Response:** Yes, the AutoPGD implementation we use is the same as the one in AutoAttack. However, we report AutoPGD results separately to provide a more fine-grained comparison of adversarial robustness across different attack strategies. While AutoAttack is generally stronger than AutoPGD, it is an ensemble attack and comes with a significantly higher computational cost. In practical adversarial scenarios, attackers with limited computational resources may prefer a cheaper attack like AutoPGD. Therefore, showing the robustness aganist AutoPGD is valuable.

---

> ### Author Response · Authors · 2025-03-17
>
> **Comment:** The adversarial accuracy under AutoPGD seems very high for ResNet18 in Table 1 (for 8/255 on CIFAR10, 63%). Other researchers can only get similar numbers with the use of unlabeled data.
>
> **Response:** We confirm that no additional unlabeled data was used in any of our experiments. To ensure reproducibility, we have shared our code, including training from scratch and evaluation against different attacks.

---

> ### Author Response · Authors · 2025-03-17
>
> **Comment:** In Table 1, why is there only one clean accuracy for CA-AT and vanilla AT under different attack budgets? The clean accuracies for these different models should be different.
>
> **Response:** We appreciate this concern. The different attack budgets reported in Table 1 do not indicate the attack budgets for adversarial examples in training but for the budget utilized for evaluation. The clean accuracy reported in Table 1 corresponds to the model trained under the default attack budget (8/255). We have modified the descriptions for Table 1 to make it more clear.

---

> ### Author Response · Authors · 2025-03-17
>
> **Comment:** Share the code and model.
>
> **Response:** We have uploaded a light version of our code with trained models (checkpoints) in the anonymous github repo (https://anonymous.4open.science/r/CA-AT-Light-TMLR), along with a detailed README file explaining how to run the experiments. Please let us know if you have any problems running this code.

---

### Review · Reviewer_qzf3 · 2025-03-08

**Summary Of Contributions:**

The paper addresses a limitation of existing adversarial training (AT) methods, which is the conflict between gradients derived from standard and adversarial losses. The authors argue that traditional weighted-average approaches to adversarial training suffer from gradient conflicts, which negatively affect the trade-off between standard accuracy and adversarial robustness. To tackle this issue, the authors propose Conflict-Aware Adversarial Training (CA-AT),  that leverages gradient projection based on cosine similarity to mitigate gradient conflict.

**Audience:**

Yes

**Claims And Evidence:**

Yes

**Requested Changes:**

- comparison with more than vanilla AT
- explain the counterintuitive results
- comment on the points above

**Strengths And Weaknesses:**

+ The paper is well written, well presented and easy to follow.
+ I do not agree that the first element of the contributions ("we shed light on the existence of conflict ...") is a novelty in essence. The conflict is a known fact that motivated all the work on tradeoff approaches between clean and robust accuracy in AT. However, the theoretical underpinning provided by Theorem 1 is interesting and supports the motivation, as it highlights the increasing gradient conflict with a higher adversarial attack budget.

- I do not quite understand some loose statements like : "... such an operation should guarantee the standard accuracy because its priority is higher adversarial accuracy".

- I expected the experimental setup to systematically compare with related work trying to find accuracy/robustness tradeoff. The authors seem to compare only with Vanilla AT

- In Table 1, it seems unexpected that the (robust) accuracy of the proposed approach under adversarial noise with a magnitude 4x higher than the budget used for AT remains sometimes almost unchanged, which is 8/255. The authors need to comment on this and explain.

- In page 10, it is not rigorous to base the statement "the conflict μ would be more serious if we utilize adversarial examples with larger attack budget δ during AT" on Theorem 1 -- Theorem 1 holds only under low noise bugdet.

- A study on the impact of the hyperparameters of the approach would help

Minor: page 6-- "gc+gc", I suppose the authors mean gc+ga

---

> ### Author Response · Authors · 2025-03-17
>
> We truly appreciate your constructive comments, and here are our responses to the proposed questions. Hope our responses can address your concerns.
>
>
> **Comment:** I do not agree that the first element of the contributions (‘we shed light on the existence of conflict …’) is a novelty in essence.
>
> **Response:** We agree that the conflict between clean and robust accuracy is a well-known motivation underlying existing approaches. However, to the best of our knowledge, no prior work has explicitly demonstrated this conflict empirically. In our paper, Figures 2 and 3 provide direct empirical evidence. If you are aware of any previous studies that have also demonstrated such a conflict empirically, we would greatly appreciate it if you let us know so that we can properly cite them.

---

> ### Author Response · Authors · 2025-03-17
>
> **Comment:** I do not quite understand some loose statements like: ‘… such an operation should guarantee the standard accuracy because its priority is higher adversarial accuracy.’
>
> **Response:** We apologize for the confusion and thank you for pointing out. We have revised the sentence to “The operation guarantees the standard accuracy the standard accuracy because its priority is higher than the adversarial accuracy.”
> This statement highlights one of the core design principles of CA-AT that standard accuracy is always prioritized above adversarial accuracy. Concretely, we implement this principle by updating the parameters using the clean gradient $g_c$ when the cosine similarity $\phi = \cos(g_c,g_a)$ is larger than the threshold $\gamma$. Please refer to Eq. (4) and Algorithm 1 in our paper for further details.

---

> ### Author Response · Authors · 2025-03-17
>
> **Comment:** Comparison with more than vanilla AT.
>
> **Response:** Thank you for this suggestion. We would like to clarify that “Vanilla AT” refers to a family of adversarial training methods (e.g., Cross Entropy, TRADES, and CLP) based on a linearly hybrid loss (Eq. (2)). In addition, we have now included results for MART [1] in Table 5 in the Appendix. These new results further demonstrate the superiority of CA-AT.
>
> [1] Improving Adversarial Robustness Requires Revisiting Misclassified Example

---

> ### Author Response · Authors · 2025-03-17
>
> **Comment:** In Table 1, it seems unexpected that the (robust) accuracy of the proposed approach under adversarial noise with a magnitude 4× higher than the budget used for AT sometimes remains almost unchanged.
>
> **Response:** This is an insightful observation and indeed highlights that CA-AT produces robust models across varying attack budgets for adversarial attacks such as PGD and FGSM. However, this trend is not always consistent. For example, for AutoAttack, the adversarial accuracy drops significantly (from 0.63 to 0.44 for ResNet18). Explaining this discrepancy is challenging, as multiple factors can influence it, including the types of adversarial attacks, model architectures, and datasets.

---

> ### Author Response · Authors · 2025-03-17
>
> **Comment:** In page 10, it is not rigorous to base the statement "the conflict $\mu$ would be more serious if we utilize adversarial examples with larger attack budget $\delta$ during AT" on Theorem 1.
>
> **Response:** Thank you for correcting this. You are correct. The $\epsilon$ used in the Taylor series expansion should be small, otherwise, the term $\mathcal{O}(\|\epsilon\|^2)$ will be too large to be neglected. We have revised the statement.

---

> ### Author Response · Authors · 2025-03-17
>
> **Comment:** A study on the impact of the hyperparameters of the approach would help.
>
> **Response:** We appreciate this suggestion. In response, we conducted additional ablation experiments on both Vanilla AT and CA-AT under different hyperparameter settings, including batch sizes (Figure 12(b)) and learning rates (Figure 12(a)) shown in the Appendix. Our findings show that while different hyperparameter choices affect standard accuracy and adversarial accuracy, CA-AT consistently achieves better performance and robustness across these different settings.

---

### Author Response · Authors · 2025-03-31

Dear reviewers,

We appreciate your feedbacks. We would like to know if you have any follow-up questions, and we will be happy to discuss them if you have.

---

### Decision · Action_Editor_6mhN · 2025-04-16

**Recommendation:** Reject

**Comment:**

This paper proposes a new method, namely Conflict-Aware Adversarial Training, to boost adversarial robustness of a DNN. However, various issues with experiments have not been well addressed, such as reliability of results in tables, tiny-image experiments and PGD attacks with increased iterations. The experiments in the paper rely heavily on synthetic or small‐scale datasets and lack a rigorous comparison with state‐of‐the‐art adaptive adversarial training methods. Additionally, the paper does not provide detailed ablation studies on key hyperparameters, nor does it offer a comprehensive analysis of the computational overhead.

Furthermore, reviewers consider the current version lacks clarity and readability. There is a lack of a comprehensive discussion on robustness overfitting and a broader impact statement. The presentation (figures, tables, and clarity in exposition) also requires significant improvement.

**Audience:**

Experts in the field of adversarial training and safefy may be interested in the finding of this paper.

**Claims And Evidence:**

This paper proposes a new method, namely Conflict-Aware Adversarial Training, to boost adversarial robustness of a DNN. However, various issues with experiments have not been well addressed, such as reliability of results in tables, tiny-image experiments and PGD attacks with increased iterations. The experiments in the paper rely heavily on synthetic or small‐scale datasets and lack a rigorous comparison with state‐of‐the‐art adaptive adversarial training methods. Additionally, the paper does not provide detailed ablation studies on key hyperparameters, nor does it offer a comprehensive analysis of the computational overhead.

Furthermore, reviewers consider the current version lacks clarity and readability. There is a lack of a comprehensive discussion on robustness overfitting and a broader impact statement. The presentation (figures, tables, and clarity in exposition) also requires significant improvement.

**Resubmission Of Major Revision:**

The authors may consider submitting a major revision at a later time.

---

> ### Author Response · Authors · 2025-05-13
>
> Dear AE,
>
> We appreciate your comments and respect your decision. We just think the concerns raised in your comments (1) reliability of results in tables (2) tiny-image experiments (3) PGD attacks with increased iterations (5) comparison to SOTA (6) detailed ablation studies on key hyperparameters were explained or addressed by the experimental results in our original draft and rebuttal. (7) results and discussion on robustness overfitting.
>
> We would appreciate it if you could double-check them, and please let us know if you have follow-up questions.